# Enigma: An Efficient Model for Deciphering Regulatory Genomics

Andrew J. Jung[1,2,3], Helen Zhu[1], Roujia Li[1], Alice J. Gao[1,2,3],
Tammy T.Y. Lau[1], Vivian S. Chu[1,2,3], Declan Lim[1,2],
Christopher B. Cole[1], Leo J. Lee[2,3*], Albi Celaj[1*], Brendan J. Frey[1,2,3*]

[1]Deep Genomics, [2]University of Toronto, [3]Vector Institute

## Abstract

Genomic sequence-to-function models have emerged as powerful tools for deciphering cis-regulatory grammar to advance our understanding of disease biology and guide therapeutic development. Recent advances have been driven by multi-task training of large transformer-based models on thousands of genome tracks. However, these performance gains have come at significant computational cost for both training and inference, hindering large-scale applications and slowing future model development. Here, rather than continuing to scale model size and add more training tracks, we focus on architectural efficiency and train on a substantially smaller, curated set of genome tracks. Our model Enigma achieves competitive performance with current state-of-the-art models at single-base resolution while substantially reducing computational cost. On zero-shot variant effect prediction benchmarks, Enigma outperforms the leading open-source model, the Borzoi ensemble, while using 10.9% of its compute and improving resolution from 32 bases to a single base. Compared to AlphaGenome, Enigma achieves 90.4 - 97.3% of its performance using 7.5% of its estimated compute. These improvements in efficiency can facilitate further development of models for regulatory genomics. We demonstrate this by fine-tuning Enigma on predicting three new molecular phenotypes (ChIP-seq, RNA half-life, and translation efficiency) achieving or exceeding the performance of state-of-the-art task-specific models. We are providing Enigma for non-commercial use to benefit the broader research field

Code: https://github.com/deepgenomics/enigma

## 1 Introduction

An important question in disease genetics is deciphering how variations in genomic sequences mediate gene regulation. With the exponential growth in functional genomics data, machine learning approaches have emerged as a promising paradigm for decoding cis-regulatory grammar in a data-driven manner (Barbadilla-Martínez et al., 2025). In particular, sequence-to-function models learn to predict genome coverage from next-generation sequencing experiments measuring transcriptional and epigenetic states from an input DNA sequence. Unlike genomic language models trained on unlabeled sequences, sequence-to-function models can predict variant and therapeutic effects in a cell-type and disease-specific manner, along with their molecular mechanisms.

Current state-of-the-art (SOTA) models, including Enformer (Avsec et al., 2021a), BigRNA (Celaj et al., 2023), Borzoi (Linder et al., 2025), and AlphaGenome (Avsec et al., 2025), use CNN-transformer hybrid architectures to capture long-range cis-regulatory interactions with context windows spanning hundreds of kilobases (up to 1 megabase (Avsec et al., 2025)) and predict thousands of tracks across multiple modalities and cell types. Notably, they have shown promise as foundation models for DNA and RNA regulation, demonstrating strong performance across downstream applications including variant effect prediction, fine-tuning

to capture new molecular properties or cell types (Lal et al., 2024; Gagneur et al.; Yuan et al., 2025), and therapeutics design (Celaj et al., 2023).

However, as these models grow more powerful, they have also become increasingly large and computationally expensive. For example, AlphaGenome was trained on a dataset of 7,058 tracks from 8 modalities, requiring 131,072 TPUv3 hours to train 64 ensemble models and 4,608 H100 hours for distillation (Avsec et al., 2025). High computational cost limits many promising applications that require large-scale inference or fine-tuning, such as interpreting variant effects genome-wide (Srivastava et al., 2025), designing nucleic acid sequences (Brixi et al., 2025), or training on personal genomes (Drusinsky et al., 2024; Rastogi et al., 2024; Spiro et al., 2025). Crucially, this further hinders future model development, especially for research groups with limited computational resources. While scaling model size and data has driven recent progress, it is worth reconsidering the balance between performance gains and computational efficiency for the field to move forward.

To address this critical need, we introduce Enigma, an efficient single-base resolution model that achieves competitive SOTA performance with substantially reduced computational cost (Figure 1). We accomplish this through innovations in both model architecture and dataset curation:

- We introduce an optimized UNet-transformer architecture that achieves single-base resolution and competitive performance with only 202M parameters, comparable in size to the current leading open-source model Borzoi (191M) and less than half of AlphaGenome (450M). Central to this architecture is FlashAttention-based self-attention (Dao et al., 2022), which addresses a key computational bottleneck in previous transformer-based genomic models.

- We curate a dataset of only 1,448 1D genome tracks across four modalities: RNA-seq coverage, DNase-seq, ATAC-seq, and novel splicing junction tracks. This represents a substantial reduction compared to Borzoi (10,219 1D tracks) and AlphaGenome (7,058 tracks across 1D and 2D representations). While recent progress has been accompanied by increasingly sophisticated track datasets, we show that competitive performance is achievable with fewer tracks and without computationally expensive 2D representations through careful dataset curation.

Enigma achieves a 2.34 times faster forward pass and 1.84 times faster backpropagation compared to a single Borzoi model, while improving output resolution from 32 bp to a single base. Compared to the Borzoi ensemble of four models, these speedups scale proportionally. Relative to AlphaGenome, Enigma requires 13.3 times fewer estimated FLOPs and reduces estimated model development costs by 24 times. Despite these efficiency gains, Enigma outperforms the Borzoi ensemble and achieves 90.4 - 97.3% performance of AlphaGenome on zero-shot variant effect prediction benchmarks. We further demonstrate that Enigma can be efficiently fine-tuned to predict novel molecular phenotypes beyond its training tracks. To benefit the broader research community, we are providing Enigma for non-commercial use.

## 2 Enigma model and dataset for efficient sequence-to-function modeling

Efficient architecture for single-base resolution. Enigma is trained to jointly predict 1,190 human and 258 mouse genome-wide tracks that measure transcriptional and epigenetic states across diverse cell types. The model is based on a hybrid UNet–transformer architecture with three main components: (1) convolutional UNet encoder blocks that capture local sequence motifs, (2) transformer blocks operating on downsampled features to model long-range interactions, and (3) UNet decoder blocks that upsample features back to single-base resolution. During training, we use 524,288 bp of input sequence context. This context window captures most of the regulatory context of genes (Tomás-Daza et al., 2023; van Arensbergen et al., 2014; Gschwind et al., 2023). We found that extending it further yields only marginal performance gains at substantially increased computational cost, consistent with ablation studies reported in (Avsec et al., 2025).

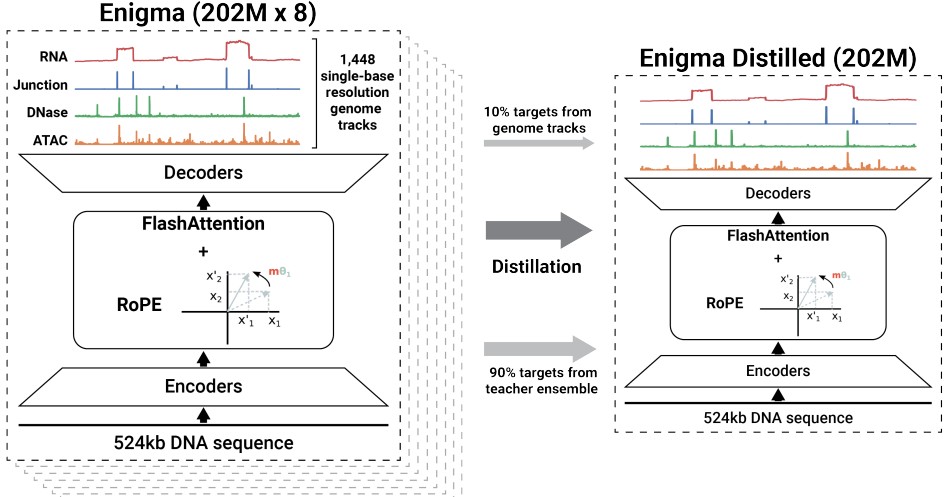

Figure 1: Overview of Enigma model, track dataset, and distillation training. Enigma is a 202M-parameter UNet–transformer hybrid model that takes 524,288 bp of DNA sequences as input and outputs single-base resolution track predictions. A core design focus of Enigma is computational efficiency while achieving competitive performance, and one key component for this is FlashAttention-based self-attention layers. The training dataset comprises 1,448 tracks across four modalities: RNA-seq, splice junction, DNase-seq, and ATAC-seq. The student model is distilled from an ensemble of eight teacher models, using a mixture of predictions from teacher models (90%) and ground truth genome tracks (10%).

We designed Enigma's architecture with computational efficiency as a primary objective, aiming to achieve competitive performance without increasing model size or architectural complexity relative to prior SOTA models. Enigma has 202M trainable parameters, similar to Borzoi's 191M parameters, yet improves the output resolution from 32 bp to a single base. Compared to AlphaGenome (450M parameters), Enigma is less than half the size and operates entirely on 1D representations, rather than the computationally demanding 2D representations that AlphaGenome employs to explicitly capture pairwise interactions.

A major computational bottleneck in transformer-based genomic models is the quadratic complexity of self-attention with respect to sequence length. Previous attempts to mitigate this issue with more efficient alternatives have generally led to degraded performance or a dependency on pre-trained weights from existing models (Hingerl et al., 2024; Holmes et al., 2025). To avoid these limitations, Enigma employs FlashAttention-based self-attention layers (Dao et al., 2022), which provide substantial speedups through optimized memory access while preserving the full representation capacity of self-attention. A key challenge is that the custom, non-sinusoidal positional encodings widely used in prior transformer-based genomic models (e.g., Enformer, BigRNA, Borzoi) are incompatible with FlashAttention. These encodings capture genomic inductive biases that help distinguish regulatory elements across a wide range of distances and are critical for good performance (Avsec et al., 2021a). Here, we adopt rotary positional encodings (RoPE) (Su et al., 2024), which are compatible with FlashAttention, and compensate for the loss of domain-specific inductive biases by (1) introducing data augmentation through random shifting of input sequences and corresponding target tracks by up to 2,048 bp, and (2) increasing regularization via higher dropout rates and weight decay. Our approach is consistent with recent work in deep learning, where more general and computationally efficient architectures can replace specialized ones with domain-specific biases through data augmentation and regularization (Abramson et al., 2024; Wang et al.; Chen et al., 2020; Steiner et al., 2021).

Beyond the transformer layers, we applied additional optimizations throughout the architecture to improve both computational efficiency and training stability (detailed in the Appendix B). One such optimization is early cropping during UNet decoding. To remove boundary artifacts from regions that lack sufficient sequence context, existing models typi-

cally crop edge regions immediately before the final output heads (Avsec et al., 2021a; Linder et al., 2025). Instead, we crop the output features of the transformer blocks, reducing unnecessary computation over long sequence dimension in subsequent layers. Overall, our efficient Enigma architecture enables training on standard hardware configurations, without specialized accelerators or complex parallelism strategies (e.g. the 8-way sequence parallelism setup on TPUs used by AlphaGenome).

**Single-base dataset from four types of tracks.** We curated a training dataset of 1,448 genome tracks across human and mouse. Rather than expanding the number of modalities, we focus on just four track types (RNA-seq coverage, DNase-seq, ATAC-seq, and splice junctions), reflecting our core design objective of prioritizing efficiency and reducing complexity. We introduced novel 1D junction tracks constructed from RNA-seq data, which summarize, at single-base resolution, the number of reads spanning splice junctions at donors and acceptors, thereby capturing splice site positions and their usage (details in the Appendix C). While RNA-seq implicitly captures many aspects of transcriptional regulation, including splicing, explicitly training on junction tracks enhances splicing predictions.

Compared to existing large multi-task sequence-to-function models, our dataset uses substantially fewer modalities and tracks. Enformer is trained on 6,956 tracks spanning ChIP, CAGE, DNase-seq, and ATAC-seq, while Borzoi expands this to 10,219 tracks by adding RNA-seq. BigRNA focuses on 3,649 post-transcriptional tracks comprising RNA-seq, RNA binding proteins, and microRNA binding sites. The current SOTA model AlphaGenome is trained on 7,058 tracks from ChIP, CAGE, PRO-cap, RNA-seq, DNase-seq, ATAC-seq, 2D chromatin contact maps, and 2D splice junction tracks. Although these additional assays do contain signals not directly observed in our four modalities, we hypothesize that much of the regulatory information for training a large, multi-task sequence-to-function model can be inferred from RNA-seq, DNase-seq, ATAC-seq, and junction tracks. To further reduce redundancy and noise from individual experiments, we merge tracks that share both cell-type ontology annotations and assay type into single representative tracks, a strategy also adopted in Avsec et al. (2025). This substantial reduction in the number of tracks is critical for making single-base resolution training practical: for example, directly scaling Borzoi's 10,219 tracks from 32 bp resolution (already requiring 2.4 TB of storage) to single-base resolution would incur prohibitive storage requirements and severe I/O bottlenecks during training.

**Distillation to improve performance while maintaining efficiency.** Model ensembling (Lakshminarayanan et al., 2017) has been shown to improve predictive performance, particularly for out-of-distribution tasks (Zhou et al., 2024; Linder et al., 2025), but typically incurs prohibitive computational cost for both inference and fine-tuning on downstream applications. To approximate the benefits of an ensemble while retaining the efficiency of a single model (Malinin et al., 2019; Zhou et al., 2024; Avsec et al., 2025), we trained a single distilled Enigma model from an ensemble of eight teacher models, using a combination of the teacher ensemble's predictions (90%) and the ground-truth genome tracks as targets (10%). We choose an ensemble size of eight to balance diminishing accuracy gains from larger ensembles against increased computational costs. For comparison, AlphaGenome used a 64-model ensemble, incurring considerably higher training cost (Section 3.1).

## 3  Results

Enigma achieves significant improvements in computational efficiency over current SOTA models while maintaining competitive predictive performance at single-base resolution. We benchmarked against Borzoi (the leading open-source model) and AlphaGenome (accessible via API), both of which have already demonstrated superior performance to Enformer across tasks; we excluded BigRNA as it is not open-sourced. Since our focus is on balancing efficiency with performance for a large multi-task sequence-to-function model, we also excluded task-specific models such as Pangolin (Zeng & Li, 2022), SpliceAI (Jaganathan et al., 2019), DeltaSplice (Evsyukova et al., 2010), and ChromBPNet (Pampari et al., 2025) (see Avsec et al. (2025) for comprehensive comparisons between AlphaGenome and these lead-

ing task-specific models). Compared to Borzoi and AlphaGenome, Enigma demonstrates substantial reductions in both training and inference costs without sacrificing accuracy.

## 3.1 Enigma substantially reduces computational costs

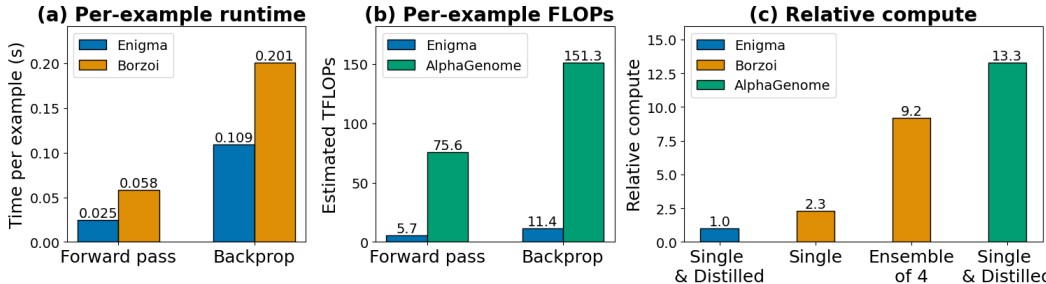

Figure 2: Computational cost of Enigma compared with Borzoi and AlphaGenome. (a) Per-example runtime (in seconds) for forward pass and backpropagation on a single NVIDIA H100 GPU for Enigma and Borzoi. (b) Estimated TFLOPs per forward pass and backpropagation for Enigma and AlphaGenome, derived from its published model architecture. (c) Relative compute for a forward pass, normalized to Enigma.

**Direct runtime comparison with Borzoi** Enigma reduces wall-clock time per example for the forward pass and backpropagation to 43% and 54% of Borzoi's, respectively (2.34 times and 1.84 times faster), when benchmarked on the same NVIDIA H100 GPU (Figure 2a, details in Appendix E.1). These per-example gains translate into substantial reductions in total training time: training Borzoi from scratch required ~25 days on two A100 40GB GPUs (Linder et al., 2025), whereas Enigma is estimated to complete training in 10 days on the same GPUs[1], a ~2.5-fold reduction in GPU hours. Notably, these efficiency gains are achieved while Enigma predicts at single–base resolution, compared to Borzoi's 32 bp resolution.

**Estimating computational efficiency versus AlphaGenome** Since AlphaGenome was, until very recently, only accessible via API, we cannot directly benchmark wall-clock runtime[2]. Instead, we approximate per-example computational cost using FLOP estimates derived from the published architecture (Avsec et al., 2025). We estimate that AlphaGenome requires approximately 75 TFLOPs per forward pass, whereas Enigma requires only 5.7 TFLOPs, corresponding to a ~13-fold reduction in compute for a single inference (Figure 2b). Assuming that backpropagation approximately doubles the FLOP cost of the forward pass, this implies a similar ~13-fold reduction in FLOPs per training step.

The efficiency gap is even more pronounced in total training cost. AlphaGenome trains 64 ensemble models on 512 TPUv3 cores for 4 hours each, followed by a three-day distillation phase on 64 H100 GPUs, totaling 131,072 TPUv3 core-hours and 4,608 H100 GPU-hours (Avsec et al., 2025). Using standard on-demand cloud pricing, this corresponds to an estimated compute cost of ~$145,000 (Appendix E.2). In contrast, Enigma trains 8 ensemble models on 4 H100 GPUs for 42 hours each, followed by an 88-hour distillation training on 8 H100 GPUs, for a total of 2,048 H100 GPU-hours (approximately $6,000). This ~24-fold reduction in training cost demonstrates that competitive performance (Section 3.2) is achievable without the prohibitive computational budget required by current large SOTA models.

---

[1]In practice, we trained Enigma on four H100 GPUs for 42 hours, which is more cost-effective.

[2]Very recently, AlphaGenome model weights were made available for non-commercial use. This section will soon be updated with a direct runtime comparison. In a preliminary comparison, Enigma achieved an approximately 10 times faster forward pass, consistent with the FLOP estimate of a 13-fold reduction in compute. More thorough benchmarking is currently underway to ensure the two models are compared in a fair setting.

Relative computational costs   Normalizing to Enigma as baseline, a single Borzoi model requires 2.3 times more compute per forward pass, while an ensemble of four Borzoi models requires 9.2 times more compute. AlphaGenome requires 13.3 times more compute based on FLOP estimates (Figure 2c).

## 3.2   Zero-shot variant effect predictions

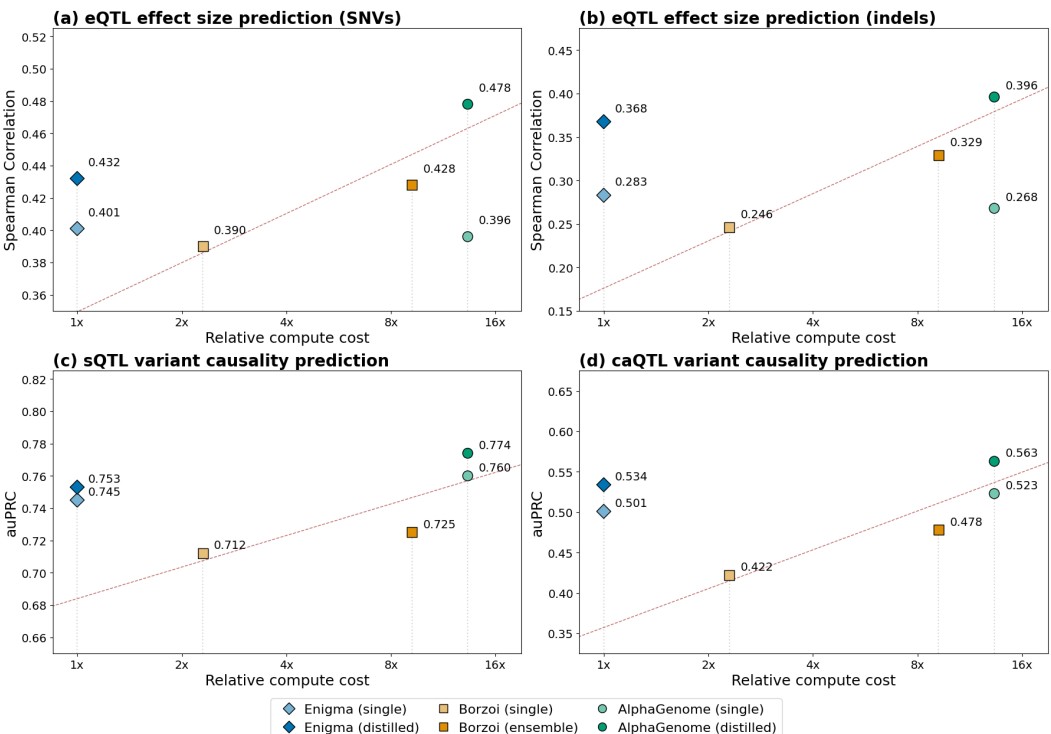

Figure 3: Zero-shot variant effect prediction performance versus relative compute cost. (a) eQTL effect size prediction for SNVs and (b) indels. (c) Causality prediction for sQTL and (d) caQTL. Models compared include Enigma (single and distilled), Borzoi (single model and ensemble of 4), and AlphaGenome (single and distilled). The dashed line shows a log-linear trend line fit to existing models achieving the best performance at each compute level, illustrating the compute cost typically required for incremental performance gains in prior approaches.

We evaluated Enigma against Borzoi and AlphaGenome on the same eQTL, sQTL, and caQTL datasets used in Linder et al. (2025); Avsec et al. (2025). We focused on zero-shot variant effect prediction tasks because the models were trained on different datasets with varying processing pipelines, making direct comparison on held-out tracks difficult. For Borzoi and AlphaGenome, we used the recommended variant scoring functions described in Linder et al. (2025); Avsec et al. (2025). For Enigma, we adopted the same scoring functions for expression and accessibility variants, and introduced a novel variant scoring function for splicing based on the maximum absolute log-fold change of junction track predictions (Appendix F).

Figure 3 summarizes the results, with the x-axis showing relative compute cost per forward pass. The distilled Enigma model consistently outperforms the leading open-source Borzoi model across all tasks, while requiring less compute and increasing resolution from 32bp to a single base. Compared to the Borzoi ensemble, which achieves competitive performance, Enigma uses only 10.9% of its compute. Relative to the more efficient Borzoi single model, Enigma remains 2.3 times more efficient while improving performance by 5.8 - 49.6%. While AlphaGenome achieves the highest overall performance, Enigma remains competitive, achieving 90.4 - 97.3% of its performance, despite using only 7.5% of its estimated compute cost. Notably, the AlphaGenome single model performs comparably to both Borzoi and

Enigma on the two eQTL tasks, suggesting that much of its advantage may derive from 64-model ensemble distillation.

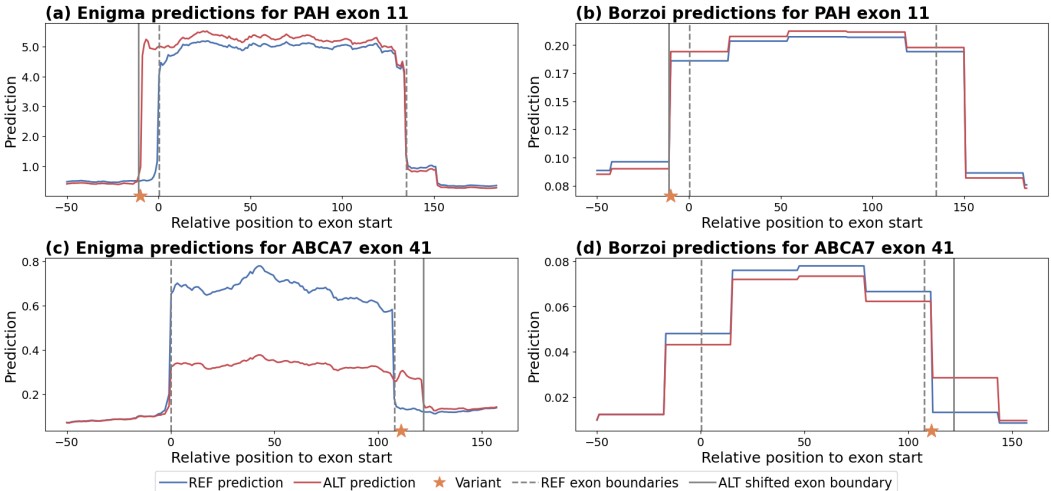

Figure 4: Enigma (single-base) and Borzoi (32 bp) predictions of RNA-seq coverage around two splice-altering variants resulting in exon extension. Blue and red denote predictions for the reference (REF) and alternative (ALT) allele sequences, orange stars mark variant positions, gray dashed lines indicate annotated exon boundaries, and gray solid lines indicate shifted exon boundaries. (a, b) Variant chr12:103237568:C:T extends PAH exon 11 by 11 bases upstream (Dworniczak et al., 1991) (c, d) Variant chr19:1061892:G:C extends ABCA7 exon 41 by 14 bases downstream, along with exon skipping (Vasquez et al., 2017; De Roeck et al., 2017) Enigma correctly captures exon boundaries for both REF and ALT, including decreased coverage from exon skipping in (c), whereas Borzoi's 32 bp resolution obscures precise exon boundaries and changes caused by these variants.

A key advantage of single-base resolution is the ability to predict subtle and complex molecular changes, particularly given that many variants and therapeutics alter regulatory events at the resolution of one or a few bases. For example, splicing aberrations can cause small exon extensions that lead to substantial changes at the protein level. Figure 4a, c illustrates that Enigma is able to pinpoint the effects of two known disease variants in PAH (Dworniczak et al., 1991) and ABCA7 (Vasquez et al., 2017; De Roeck et al., 2017) that cause exon extensions of 11 and 14 bases, respectively, with predicted RNA-seq coverage dropping sharply at the expected exon boundaries. Additionally, we accurately predict that the ABCA7 variant leads to substantial exon skipping (De Roeck et al., 2017), reflected by reduced predicted RNA-seq coverage across the skipped exon (Figure 4c). In contrast, Borzoi's 32 bp resolution obscures the precise exon boundaries and changes caused by these variants (Figure 4b, d). Together, these examples demonstrate how Enigma's single-base predictions can support mechanistic interpretation by revealing not only whether a variant affects splicing, but also the precise nature of the resulting splicing alterations.

### 3.3 Enigma enables efficient fine-tuning on downstream tasks

There has been significant interest in applying large genomic sequence-to-function models to a wide range of downstream tasks. While approaches like parameter-efficient fine-tuning have been proposed to reduce computational costs, full fine-tuning has been shown to achieve the best performance (Yuan et al., 2025). Here, we leverage Enigma's efficiency to fine-tune it on three prediction tasks.

ChIP-seq. Although Enigma was trained without ChIP-seq tracks for computational efficiency, here we show that it can be effectively adapted to predict transcription factor (TF) binding sites and histone modifications through fine-tuning. We added a ChIP-seq prediction head to the pre-trained Enigma and compared two fine-tuning approaches: (1) training only the prediction head while freezing the rest of the model, and (2) full fine-tuning.

After only 3 epochs of training on the same ChIP-seq tracks used by Borzoi, both approaches achieved competitive performance (Table 1). Full fine-tuning reaches near-parity with Borzoi (Pearson correlation of 0.590 vs. 0.595) while head-only fine-tuning achieves correlation of 0.520, consistent with the previous findings that full fine-tuning often achieves the best performance (Yuan et al., 2025). These results demonstrate that Enigma's learned representations can be effectively adapted to predict new regulatory genomics tracks, despite their absence during pre-training.

Table 1: Fine-tuning performance on ChIP-seq tracks from the Borzoi dataset. Pearson correlations were computed on held-out tracks (the same 'fold3' test set used in Borzoi). Values show mean correlations across four model replicates.

|  | Borzoi | Enigma (full) | Enigma (head only) |
|---|---|---|---|
| Pearson R | 0.595 | 0.590 | 0.520 |

RNA half-life and translation efficiency. Through pre-training on RNA-seq data, Enigma has likely learned latent representations relevant to various RNA properties. Here, we demonstrate that Enigma can be fine-tuned to predict RNA half-life and translation efficiency.

We fine-tuned Enigma using the same datasets as Saluki (Agarwal & Kelley, 2022) for RNA half-life prediction and RiboNN (Zheng et al., 2025) for translation efficiency prediction, following their respective training and evaluation setups. Notably, while both Saluki and RiboNN relied on additional input channels encoding genomic annotations, such as splice sites and codon reading frames, and aligning open reading frame to boost performance, Enigma uses only RNA sequences as input. For each task, we added a task-specific head which takes pooled embeddings from Enigma as input.

As shown in Table 2, Enigma achieves performance comparable to both Saluki and RiboNN, despite using only sequences as input. Notably, Enigma significantly outperforms sequence-only versions of these models.

Table 2: Performance on RNA property prediction tasks. Pearson correlation coefficients for human held-out test sets are reported. Models marked with '(+annotations)' use sequence plus genomic annotations, such as the first reading frame of each codon and 5' splice site. Unmarked models use sequence only.

|  | RNA half-life |
|---|---|
| Saluki | 0.62 |
| Saluki (+annotations) | 0.77 |
| Enigma | **0.79** |

|  | Translation efficiency |
|---|---|
| RiboNN | 0.66 |
| RiboNN (+annotations) | 0.71 |
| Enigma | **0.73** |

## 4   Discussion

Enigma demonstrates near state-of-the-art performance on genomic sequence-to-function predictions at substantially reduced computational cost. This reflects our core design philosophy: while improving predictive accuracy is important, marginal gains may not justify orders of magnitude increases in computation, especially when such costs limit practical application and subsequent model development. Guided by this principle, we prioritized balancing accuracy with efficiency, outperforming the leading open-source model while substantially lowering resource requirements and predicting at single-base resolution. Compared to AlphaGenome, Enigma achieves 90.4% to 97.3% of the performance, a point at which significant gains in efficiency may outweigh modest reductions in accuracy, particularly for large-scale applications or research groups with limited computational resources.

Since AlphaGenome is not directly accessible, we used estimated FLOPs as a proxy for true computational cost. It is possible that AlphaGenome employs custom kernels to optimize efficiency, which would narrow the gap reported here. However, Enigma achieves its efficiency gains entirely through readily accessible open-source tools, without specialized kernel designs or hardware configurations that may be unavailable to most research groups.

This work suggests several directions for follow-up investigation. In addition to more extensive evaluations across tasks and datasets, an important direction is to systematically ablate the contributions of 1D versus 2D tracks. In this study, we focus on 1D representations for computational efficiency and demonstrate that strong performance can be achieved without explicit 2D representations. However, 2D tracks may capture additional structural and long-range contextual signals that are not easily recoverable from 1D tracks alone. A natural next step is to quantify the incremental benefit of the 2D junction tracks and contact maps and to explore strategies for incorporating them while keeping computational costs manageable, such as by introducing 2D representations only at the output layer and fine-tuning for a limited number of epochs.

Finally, despite substantial progress in sequence-to-function modeling, current models still struggle to accurately capture cell-type specific regulatory context, differential splicing, long-range distal regulatory interactions, and predicting expression from personal genomes. Addressing these challenges will likely require significant additional development, potentially including specialized fine-tuning strategies, retraining new models from scratch, or integrating sequence-to-function models with cell models. We anticipate that compute-efficient base models, such as Enigma, can accelerate this progress by lowering the barrier to iterative experimentation. To this end, we have made Enigma available to the research community for non-commercial use.

Acknowledgments

We thank Caitlin F. Harrigan and Leonardo Cotta for insightful discussions. We also acknowledge Vector Institute for additional compute resources.

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

## A   Appendix

## B   Model architecture

Enigma is based on a hybrid UNet-transformer architecture with three main components: (1) UNet encoder blocks, (2) a stack of transformer layers, and (3) UNet decoder blocks. Given an input sequence, the model jointly predicts 1,190 human and 258 mouse genome tracks at single-base resolution.

The core design philosophy underlying Enigma prioritizes computational efficiency while achieving competitive performance, without scaling model size or introducing additional architectural complexity. Enigma has 202M trainable parameters — comparable to Borzoi (191M) and less than half of AlphaGenome (450M). Relative to Borzoi, Enigma improves the output resolution from 32 bp to a single base despite similar size. Compared to AlphaGenome, Enigma operates entirely on 1D representations rather than the computationally demanding 2D representations that AlphaGenome employs to explicitly capture pairwise interactions.

UNet encoder blocks. During training, Enigma uses an input context of 524,288 bp — at inference time, the model supports arbitrary sequence lengths as long as they fit into GPU VRAM. The input sequence is one-hot encoded and projected into a 256-dimensional embedding by a 1D convolutional layer with kernel size 15.

The sequence embedding is then downsampled from 1 bp to 128 bp resolution through 7 UNet encoder blocks, while the embedding dimension increasing from 256 to 1,536. Each encoder block consists of max pooling (kernel size 2, stride 2) followed by a single 1D convolution (kernel size 5). We use GELU activations Hendrycks & Gimpel (2016) and Group Normalization Wu & He (2018) with a group size of 32. Compared to Batch Normalization

Ioffe & Szegedy (2015) used in previous CNN-transformer models, Group Normalization can improve stability, particularly with small batch sizes.

Compared to prior CNN–transformer sequence-to-function models, we intentionally simplify the UNet encoder to reduce early-stage compute, where sequence lengths are longest. Specifically, Enigma's encoder blocks contains only a single convolutional layer (rather than two-layer residual blocks), fewer parameters overall, and a smaller initial embedding dimension (256 vs. 512 in Borzoi and 768 in Enformer, BigRNA, and AlphaGenome). This design choice reduces FLOPs in the high-resolution stages of the model, without sacrificing performance given that transformer blocks provide the bulk of representational capacity in UNet-transformer models.

Transformer layers. After the encoder blocks, 8 transformer blocks process the 1,536-dimensional embeddings. A major computational bottleneck in transformer-based genomic models is the quadratic complexity of self-attention with respect to sequence length. To improve efficiency without sacrificing the representational capacity of self-attention[3], Enigma employs FlashAttention, which provides substantial speedups through optimized memory access.

Since the custom positional encodings used in prior genomic transformer models (e.g., Enformer, BigRNA, Borzoi) are incompatible with FlashAttention, we adopt rotary positional encodings (RoPE) with a base frequency of 10,000. To compensate for the loss of domain-specific inductive biases provided by the custom encodings, we (1) use data augmentation through random shifting of input sequences and corresponding target tracks by up to 2,048 bp, and (2) increase regularization via higher dropout rates and weight decay. Dropout is applied to the attention, within MLP blocks, and to the outputs of both attention and MLP blocks before they are added to the residual connections. We use dropout rates of 0.2 for attention and 0.3 elsewhere. We use multi-head attention with head dimension of 192, MLP expansion factor of 2, GeLU activations for the MLP blocks, and Layer Normalizations Ba et al. (2016). We use QK-norm Henry et al. (2020) to stabilize training, which we found particularly help when training with noisy single-base resolution track data.

UNet decoder blocks. After the transformer layers, 7 UNet decoder blocks upsample the sequence embedding from 128 bp to 1 bp resolution while decreasing the embedding dimension from 1,536 to 768. Each decoder block consists of upsampling by a factor of 2, a pointwise linear layer to reduce the embedding dimension, addition of the UNet skip-connection, and a separable convolution (kernel size 3). The skip-connection is projected to the reduced embedding dimension before addition. Decoder blocks use the same activations and normalizations as the encoders.

One key optimization is early cropping during UNet decoding, leveraging the fact that edge regions are often cropped before computing the loss to remove boundary artifacts due to insufficient sequence context Avsec et al. (2021a); Linder et al. (2025). We crop the sequence embedding immediately after the transformer blocks, reducing unnecessary computation along the sequence dimension in subsequent layers. Furthermore, decreasing the embedding dimension as sequence resolution increases reduces FLOPs in the high-resolution stages, mirroring the efficiency focus in the encoders. This contrasts with Borzoi, which maintains a constant embedding dimension of 1,536 to support a larger number of output tracks (10,219 track outputs versus Enigma's 1,448).

After the final decoder block, an MLP head (with GELU activation and dropout of 0.1) processes the single-base resolution sequence embedding to predict track output. Softplus activation is applied to ensure non-negative values.

---

[3]Alternatives such as state-space models (SSMs) may sacrifice representational capacity; SSM-based architectures often incorporate self-attention layers to achieve competitive performance Lieber et al. (2024); Brixi et al. (2025)

## C   Data processing

We processed RNA-seq, DNase-seq, and ATAC-seq datasets from ENCODE Consortium et al. (2012) and GTEx Consortium (2020), resulting in a total of 1,448 single-base resolution genome tracks across human and mouse.

For RNA-seq, we downloaded processed bigWig tracks from ENCODE and GTEx (via Recount3 Wilks et al. (2021)). For DNase-seq and ATAC-seq, we convert ENCODE BAM files into bigWig tracks using a custom script provided by the authors of ChromBPNet Pampari et al. (2025). This script applies enzyme-specific cut bias correction, following the procedure described in Avsec et al. (2025).

All bigWig tracks are normalized to 100 reads per million (RPM) to account for differences in sequencing depth and read length. We then average tracks belonging to the same cell-type ontology term and assay type, resulting in a single representative track per biological context. This reduces redundancy and number of tracks, improving computational efficiency and lowering storage requirements.

To obtain splice-junction tracks, we started from the aligned BAM files provided by GTEx (v8), which were generated using the STAR aligner with two-pass mode (GTEx pipeline reference). From each BAM file, we kept only uniquely mapped reads, and extracted junction counts directly from the CIGAR strings.

For each alignment, intron spanning reads were identified and the left and right overhangs flanking each junction were recorded. Junctions were kept only if both overhangs were at least 3 nucleotides (min_overhang = 3) and if they had at least one supporting read (min_reads = 1). Counts reflect the number of reads supporting a given splice junction.

For each sample, this workflow produces a distribution of supporting read counts across all splice junctions detected in that sample. To obtain tissue-level summaries, we aggregated the sample-specific junction tracks across all GTEx donors for each tissue. Mean counts per junction were computed across samples for the final tissue-level junction tracks.

After all of the processing steps, we obtain 1,190 human and 258 mouse genome tracks. Table 3 summarizes the number of tracks per species and assay type.

Table 3: Number of genome tracks by species and assay type.

| Assay type | Human | Mouse | Total |
|---|---|---|---|
| RNA-seq | 667 | 173 | 840 |
| DNase-seq | 302 | 67 | 369 |
| ATAC-seq | 167 | 18 | 185 |
| Junction (RNA-seq) | 54 | 0 | 54 |
| Total | 1,190 | 258 | 1,448 |

From the processed bigWig tracks, we create a dataset consisting of 55,497 human and 49,369 mouse examples consisting of sequence and track pairs.

We use the same set of 524,288bp sequence contigs from Borzoi, following the same 8 fold splits keeping fold4 for validation and fold3 for test. For each sequence contig, we use the center 196,608bp for the track target.

Each example is stored as NPZ files. Sequence is stored as sequence intervals and dynamically fetched during loading using GenomeKit[4]. Dynamic range of genome tracks are often very large, causing challenges in training stability. As in the previous works Linder et al. (2025); Avsec et al. (2025), we apply scaling transformations to reduce dynamic range of the track values. Track values are stored in float16. This results in a total of 104,866 number of examples with approximately 2.4TB of storage.

---

[4]https://github.com/deepgenomics/GenomeKit

## D  Model training

Enigma is trained using the Poisson-multinomial loss function Avsec et al. (2021b), following the approach of Borzoi and upweighting the multinomial loss term by a factor of 5. The learning rate increases linearly to 0.0001 during a warm-up period of 20,000 steps, then decays to 0.00001 following a cosine schedule. We use the AdamW optimizer Loshchilov & Hutter (2017) with default hyperparameters and weight decay of 0.01. During training, we apply data augmentation through random cropping and reverse complementation of input sequences and target tracks.

For training a single Enigma model to compare against a single Borzoi model and a single AlphaGenome model trained on the same training split as Borzoi (named fold3 in the API), we also use the same data splits: fold3 for testing and fold4 for validation. We train the model for 200,000 steps on 4 H100 GPUs with a batch size of 2.

For ensemble training, we use all folds following the approach of AlphaGenome, training models for 260,000 steps on 4 H100 GPUs with a batch size of 2. We train 8 ensemble models, saving the 4 checkpoints with the lowest training loss from each.

For distillation training, we train on 8 GPUs, loading 4 teacher models on each and ensuring that no two checkpoints from the same training run are loaded together. In addition to random cropping and reverse complementation, we introduce random substitutions and indels, similar to the approach of Avsec et al. (2025). We train on teacher model predictions 90% of the time and on real track data for the remaining 10%.

## E  Benchmarking computational efficiencies

### E.1  Direct runtime comparison with Borzoi

We compare the runtime of Enigma and Borzoi by measuring per-example wall-clock time for the forward and backward passes on the same NVIDIA H100 GPU. We aim to use settings that reflect realistic uses of both models.

Enigma is trained with bfloat16 mixed precision and uses the same bfloat16 mixed precision at inference. In contrast, we observe that Borzoi's performance degrades with mixed-precision inference, so we run it in float32. We set the internal precision of float32 matrix multiplications via torch.set_float32_matmul_precision('high'), which provides faster compute with no observable performance degradation compared to 'highest'. This yields a larger speedup for Borzoi running in float32 than for Enigma.

To better reflect performance in practical settings, we use real data when measuring time for backpropagation. After 20 warm-up iterations, we average the per-example wall-clock time over 100 iterations with batch size 1. We measure the time for the forward pass and backpropagation, excluding data loading, to isolate model compute from other contributing factors such as I/O speed.

### E.2  Estimating training costs

A single Enigma model took approximately 42 hours to train on 4 NVIDIA H100 GPUs, and distillation from an ensemble of 8 models took approximately 88 hours on 8 NVIDIA H100 GPUs. To estimate training costs of AlphaGenome, we refer to training durations provided in Avsec et al. (2025) (supplementary sections on 'Pretraining' and 'Distillation'). A single AlphaGenome model took approximately 4 hours on 512 TPUv3 cores, and distillation from an ensemble of 64 pretrained models took approximately 3 days on 64 NVIDIA H100 GPUs. Table 4 summarizes these computational requirements. Total hardware hours for single model training reflect the computational cost of training all models in the ensemble (8 models for Enigma, 64 models for AlphaGenome).

The total computational requirement to train the Enigma distilled model (including ensemble training) is 2,048 H100 hours, which is less than half of the 4,608 H100 hours required

Table 4: Training computational requirements for Enigma and AlphaGenome

| Model | Training Stage | Duration | Hardware | Total Hardware Hours |
|---|---|---|---|---|
| Enigma | Single model* | 42 hours | 4 H100 GPUs | 1,344 H100 hours |
| Enigma | Distillation | 88 hours | 8 H100 GPUs | 704 H100 hours |
| AlphaGenome | Single model* | 4 hours | 512 TPUv3 cores | 131,072 TPUv3 core hours |
| AlphaGenome | Distillation | 72 hours | 64 H100 GPUs | 4,608 H100 hours |

*Total hardware hours include training all models in the ensemble.

for AlphaGenome's distillation stage alone, excluding the substantial TPU costs for training its 64 ensemble.

To estimate dollar costs, we use current pricing from major cloud service providers as of November 2025. For H100 GPUs, we use $3.00 per GPU-hour, based on rates from providers such as Lambda Labs, which offers on-demand 8×H100 instances at $2.99 per H100-hour. For TPUv3, we use Google Cloud Platform's pricing of $2.00 per chip-hour. Since each TPUv3 chip contains two cores, this translates to $1.00 per core-hour. Table 5 summarizes the estimated training costs for each stage. Overall, Enigma achieves a 24-fold reduction in training costs compared to AlphaGenome ($6,144 versus $144,896), making model development more accessible to research groups with limited computational budgets.

Table 5: Estimated training costs for Enigma and AlphaGenome

| Model | Training Stage | Hardware Hours | Estimated Cost (USD) |
|---|---|---|---|
| Enigma | Single model ensemble | 1,344 H100 hours | $4,032 |
| Enigma | Distillation | 704 H100 hours | $2,112 |
| Enigma | Total | — | $6,144 |
| AlphaGenome | Single model ensemble | 131,072 TPUv3 core hours | $131,072 |
| AlphaGenome | Distillation | 4,608 H100 hours | $13,824 |
| AlphaGenome | Total | — | $144,896 |

Costs estimated using $3.00/H100-hour and $1.00/TPUv3 core-hour based on November 2025 cloud provider rates.

## F  Zero-shot variant effect predictions

We use the GTEx eQTL and sQTL datasets processed in Linder et al. (2025) and the caQTL African dataset from Pampari et al. (2025).

For evaluating AlphaGenome, we use its recommended variant scorers. For Borzoi, we use the variant scorers described in Linder et al. (2025) for SNV eQTL and sQTL, and adopt AlphaGenome's recommended scorers for indel eQTL and caQTL.

For Enigma, we use the same scorers as Borzoi and AlphaGenome for SNV eQTL, indel eQTL, and caQTL. For sQTL, we introduce a new scoring function that takes the maximum absolute log-fold change of junction predictions between reference and variant alleles.

## G  Downstream fine-tuning

### G.1  ChIP-seq fine-tuning

To adapt Enigma for predicting ChIP-seq tracks, we add a new ChIP-seq predicting head and fine-tune on ChIP-seq tracks processed and shared by the authors of Borzoi. We use the same MLP architecture, consisting of a linear layer, dropout, GELU activation, and the final output linear layer, used by the other track heads. We use the same hidden dimension of 1,280 as the other heads.

### G.2 RNA half-life and translation efficiency fine-tuning

For RNA half-life prediction, we reimplemented the Saluki model in PyTorch using the dataset provided by the authors, training models with and without genomic annotations. For translation efficiency prediction, we used the pretrained RiboNN model provided by the authors.

To fine-tune Enigma on these tasks, we appended a single-layer MLP head along with pooled embeddings formed by concatenating mean- and max-pooled representations along the sequence dimension.

