# OpenReview forum: "Enigma: An Efficient Model for Deciphering Regulatory Genomics"
_ICLR.cc/2026/Workshop/LMRL — ICLR 2026 Workshop LMRL Poster_

### Official Review · Reviewer_pY31 · 2026-02-22

**Rating:** 6
**Confidence:** 4

**Review:**

Pros:
- The paper addresses an important and timely problem in regulatory genomics by focusing on reducing computational cost while maintaining competitive predictive accuracy, which could broaden accessibility for research groups with limited resources.
- The proposed architecture design is well motivated, combining UNet encoders, transformer layers, and FlashAttention to reduce bottlenecks while maintaining representational capacity, and the description of architectural components is detailed and reproducible.
- The empirical results demonstrate a compelling efficiency–performance trade-off, showing substantial reductions in FLOPs and training cost while achieving performance close to or exceeding strong baselines on variant effect prediction tasks.


Cons:
- The methodological novelty is somewhat limited because the core contributions mainly consist of integrating existing techniques such as FlashAttention, RoPE, distillation, and architectural simplifications rather than introducing fundamentally new modeling mechanisms.
- Some key comparisons, particularly with AlphaGenome, rely on estimated FLOPs and indirect benchmarking instead of fully controlled runtime experiments, which weakens the strength of the empirical claims.
- The evaluation focuses heavily on efficiency metrics and zero-shot benchmarks, but provides limited analysis of biological interpretability or new biological insights derived from the model predictions.
- The dataset reduction strategy, while efficient, raises questions about whether important regulatory signals from omitted modalities (e.g., ChIP-seq or 2D contact maps) are lost, and the paper does not include ablations to quantify this trade-off.
- The paper lacks deeper ablation studies isolating the contributions of key design choices such as FlashAttention, RoPE, early cropping, and track curation, making it difficult to assess which components drive the performance gains.
- The statistical significance of improvements over baselines is not clearly reported, and confidence intervals or hypothesis testing would strengthen the empirical evaluation.
- The biological downstream evaluation is relatively limited in scope compared to the breadth of claims about enabling large-scale applications, and additional real-world case studies would strengthen the impact.

---

### Official Review · Reviewer_1ZPc · 2026-02-23

**Rating:** 8
**Confidence:** 3

**Review:**

This paper introduces Enigma, an efficient 202M-parameter hybrid UNet–Transformer model for regulatory genomics that achieves near state-of-the-art performance while reducing computational cost. Compared to large-scale models such as AlphaGenome and Borzoi, Enigma operates entirely on 1D representations and incorporates FlashAttention-based self-attention to significantly lower training and inference costs. With using 7.5% of computation, it attains 90–97% of AlphaGenome’s performance and outperforms Borzoi on zero-shot variant effect prediction, while also demonstrating strong fine-tuning capability on new molecular phenotypes. Although the methodological novelty is moderate and some comparisons rely on estimated FLOPs rather than direct benchmarking, the work presents a well-executed and practically valuable contribution to efficient sequence-to-function modeling, and this work would be interesting to the LMRL community.

---

### Official Review · Reviewer_8Lgn · 2026-02-24
**Clever approach to more efficient sequence-to-function modeling**

**Rating:** 7
**Confidence:** 3

**Review:**

The authors address an important problem in sequence-to-function modeling: the computationally expensive training process. Instead of scaling up a model for substantial performance gains, the authors focus on making models more efficient through thoughtful data curation and architectural advancements. Ultimately, the authors claim an order-of-magnitude efficiency gain over state-of-the-art models, while achieving comparable performance. Given the importance of efficient model training for sustainability and cost-effectiveness, I would argue that this work is worthy of acceptance at the workshop.

The authors state: “To further reduce redundancy and noise from individual experiments, we merge tracks that share both cell-type ontology annotations and assay type into single representative tracks…” Could the variability between tracks sharing cell type and assay be biologically meaningful? For example, do you control for (or stratify by) donor phenotypes, developmental stage of the cells, and/or other covariates before merging? With this said, the benefit of combining tracks to avoid technical bottlenecks is appreciated.

While the paper makes a convincing argument about Enigma’s training efficiency, the evaluation of zero-shot model performance could be expanded. For example, the authors report performance on eQTL, sQTL, and caQTL datasets without noting what these datasets represent. Although the reader can follow the included citations, the paper would benefit from acronym specifications or a brief introduction.

The authors note that “many variants and therapeutics alter regulatory events at the resolution of one or two bases” but only show two anecdotal cases. Although Figure 4 is convincing, a few more examples could support the generalizability of the result.

The authors train their model on four track types (RNA-seq coverage, DNase-seq, ATAC-seq, and splice junctions), then claim effective fine-tuning on ChIP-seq data. Which ChIP-seq tracks were used for fine-tuning? Are the ChIP-seq tracks correlated with the four tracks used during training? The paper would benefit from additional details on the ChIP-seq evaluation.

The evaluation metrics reported in Tables 1 and 2 would benefit from ranges or confidence intervals.

Please provide a Meaningfulness Statement, in accordance with the workshop instructions.

---

### Meta-Review · Area_Chair_AzZy · 2026-02-27

**Recommendation:** Accept (Poster)
**Confidence:** 5

**Metareview:**

Accept.

---

### Decision · Program_Chairs · 2026-03-02

**Decision:**

Accept (Poster)

**Comment:**

Please see the meta-review.